# Associations of area-level deprivation with adverse obstetric and perinatal outcomes in Bavaria, Germany: Results from a cross-sectional study

**Andreas Beyerlein**[1]*, **Nicholas Lack**[2], **Werner Maier**[3]

**1** Institute of Computational Biology, Helmholtz Zentrum München, Neuherberg, Germany, **2** German Bavarian Quality Assurance Institute for Medical Care, Munich, Germany, **3** Institute of Health Economics and Health Care Management, Helmholtz Zentrum München, Neuherberg, Germany

\* andreas.beyerlein@tum.de

## Abstract

### Background

We investigated associations of area-level deprivation with obstetric and perinatal outcomes in a large population-based routine dataset.

### Methods

We used the data of n = 827,105 deliveries who were born in hospitals between 2009 to 2016 in Bavaria, Germany. The Bavarian Index of Multiple Deprivation (BIMD) on district level was assigned to each mother by the zip code of her residential address. We calculated odds ratios (ORs) with 95% confidence intervals (CIs) for preterm deliveries, Caesarian sections (CS), stillbirths, small for gestational age (SGA) births and low 5-minute Apgar scores by BIMD quintiles with and without adjustment for potential confounders.

### Results

We observed a significantly increased risk for preterm deliveries in mothers from the most deprived compared to the least deprived districts (e.g. OR [95% CI] for highest compared to lowest deprivation quintile: 1.06 [1.03, 1.09]) in adjusted analyses. Increased deprivation was also associated with higher SGA and secondary CS rates, but with lower proportions of stillbirths, primary CS and low Apgar scores. When one large clinic with an unusually high stillbirth rate was excluded, the association of BIMD with stillbirths was attenuated and almost disappeared.

### Conclusions

We found that area-level deprivation in Bavaria was positively associated with preterm and SGA births, confirming previous studies. In contrast, the finding of an inverse association between deprivation and both stillbirth rates and low Apgar score came somewhat

**Data Availability Statement:** To use de-identified data, permission must be obtained from the Bavarian institute for quality assurance ('BAQ', Prof. Dr. Hermanek, mail@baq-bayern.de). The

data is expressly gathered for purposes of quality improvement of obstetrics and neonatology care. The BAQ steering committee must assert that any request for third party usage of the data is in line with the general BAQ objectives. The BAQ is bound by corresponding contracts with its participating hospitals to guarantee adherence to this protocol. The analysis code including all relevant dataset and variable names is available at https://osf.io/mc3te/.

**Funding:** The authors received no specific funding for this work.

**Competing interests:** The authors have declared that no competing interests exist.

**Abbreviations:** BAQ, Bayerische Arbeitsgemeinschaft für Qualitätssicherung; BIMD, Bavarian Index of Multiple Deprivation; CI, confidence interval; CS, Caesarean section; OR, odds ratio; SES, socioeconomic status; SGA, small for gestational age.

surprising. However, we conclude that the stillbirths finding is spurious and reflects regional bias due to a clinic which seems to specialize in termination of pregnancies.

## Background

A large body of the literature has shown that individual socioeconomic status (SES) is associated with various health outcomes [1], and there is evidence that low maternal SES is a potential risk factor for adverse perinatal outcomes [2–4]. However, a number of studies indicated that also area-level deprivation is associated with adverse obstetric and perinatal outcomes [5], e. g. gestational diabetes, Caesarian section (CS), preterm delivery, stillbirths or small for gestational age births [6–11]. All these outcomes contain major risks for mother and child. CS, for example, is a surgical intervention, but contains major corresponding risks for the mother (e.g. bleeding). Nevertheless, CS rates have increased worldwide in the last two decades [7]. Preterm births, accounting for approximately 10% of all deliveries worldwide, seem to increase in most countries, and prematurity, being the leading cause of neonatal mortality, has now also become the leading cause of childhood mortality up to age five years [12]. The worldwide number of annual stillbirths has decreased considerably since 1990, but with about two million in 2015 it is still high [13, 14].

Area-level deprivation research may consider the proportion of deprived people in an area ('compositional meaning'), the presence of possible area effects beyond the local concentration of deprived people ('collective meaning'), and the lack of infrastructural facilities or other area features ('environmental meaning') [15]. Area-level deprivation is often measured with standardised composite deprivation indices describing a structural lack of material and social resources in an area, considering e. g. income and occupation but also municipal/district revenues and environmental indicators. Therefore, these indices are valuable instruments for the analysis of area effects on health beyond the individual level, even when they might be used as a surrogate when individual socioeconomic data are not available [16].

The objective of our study was to assess the association between area-level deprivation, using an established regional area deprivation index, the Bavarian Index of Multiple Deprivation (BIMD) [17], and adverse obstetric and perinatal outcomes. Based on an established British method [15], the BIMD is a multidimensional construct combining standardised and transformed indicators in specific deprivation domains which are weighted and combined in an overall deprivation index. The BIMD was originally developed as a model for the development of the German Index of Multiple Deprivation (GIMD), being today part of a number of Indices of Multiple Deprivation (IMD) for Germany and some of its federal states. More on the methodology of the German IMD, the area-level distribution of deprivation in Germany and its association with health can be found elsewhere [17–20].

For our study, we analyzed a large population-based dataset, covering more than 800,000 births in Bavaria, Germany, from 2009 to 2016, and including a number of different and well-reported pregnancy outcomes as well as the residential address of the mother.

## Materials and methods

We used data which are regularly collected by the 'Bayerische Arbeitsgemeinschaft für Qualitätssicherung' (BAQ; German Bavarian Quality Assurance Institute for Medical Care) for national benchmarking of obstetric units in terms of clinical performance. As previously described [21, 22], these data contain anonymized information on perinatal outcomes such as

perinatal mortality or sectio as well as on maternal characteristics such as age, weight, height, diabetes status and the postal code of the mother's residential address.

As in a previous publication based on the BAQ data [6], we assigned the BIMD to the residential postal code of each mother. Based on data from official statistics, the BIMD includes seven domains of deprivation (income (weight on total BIMD: 25%), employment (25%), education (15%), municipal/district revenue (15%), social capital (10%), environment (5%), security (5%)) [17] for all 96 Bavarian districts ('Kreise', 71 rural and 25 urban districts with a minimum of 40.7 thousand, a maximum of 1.46 million and a median size of 115.6 thousand inhabitants as per 31 December 2016; https://www.regionalstatistik.de/genesis/online/). The seven domains are combined in a composite index, the BIMD. We categorized the districts by BIMD quintiles, with the first quintile (Q1) designating the least deprived and the fifth quintile (Q5) the most deprived areas.

We used the data of all n = 827,105 deliveries recorded in the BAQ database from 2009 to 2016 to which the BIMD on district level could be assigned. Stillbirths were only recorded in the BAQ database if they had a minimum birth weight of 500 g. We used logistic regression to calculate odds ratios (ORs) with 95% confidence intervals (CIs) for preterm deliveries (<37 gestational weeks), CS, stillbirths, small for gestational age (SGA) births (defined as the lower 10% of German birthweight percentiles specific for sex and gestational age [23]) and low 5-minute Apgar scores (<7) as a measure of the new-born's condition [24] by quintiles of the composite BIMD (reference: Q1, least deprived areas). Additionally, we assessed odds ratios by a linear increase in BIMD ('overall trend'), both for the composite index and for all domains. The models were calculated both unadjusted and adjusted for offspring's sex, multiple delivery, maternal age > 35 years, diabetes during pregnancy, maternal overweight (body mass index >25 kg/m$^2$) in early pregnancy, excessive gestational weight gain (according to Institute of Medicine criteria [25]), migration background, single mother status, parity, maternal smoking during pregnancy, substandard use of antenatal care (i.e. less than one antenatal visit per four weeks of gestation [26]), living in a city (>100,000 inhabitants), and year of birth. As birth mode is known to be associated with Apgar score [27], we additionally adjusted the Apgar score analyses for birth mode (vaginal birth, CS or vaginal surgery) in a sensitivity analysis. Finally, we investigated associations of BIMD with primary and secondary CS, respectively.

Statistical analyses were performed with SAS 9.4 (SAS Institute Inc., Cary, NC, USA). The analysis code is available at https://osf.io/mc3te/. A significance level of 5% was used throughout the study without adjustment for multiple testing. Maps were drawn using the open source Geographic Information System QGIS 2.18 (https://www.qgis.org/en/site/).

Ethics approval or informed consent of patients were not necessary because this was a secondary analysis of anonymous routine data [28].

## Results

Descriptive statistics of the BIMD (composite index) and all potential confounding factors are presented in Table 1, prevalences of the outcome variables by BIMD quintiles in Table 2. Missing covariate data occurred for n = 28,419 observations (13.9%) in Q1, n = 45,553 (17.7%) in Q2, n = 23,108 (19.4%) in Q3, n = 24,634 (21.2%) in Q4 and n = 26,301 (20.2%) in Q5, respectively. In pregnancies with missing covariate data, slightly higher prevalences were observed for all outcomes (preterm delivery: 9.1%, stillbirth: 0.4%, CS: 35.9%, SGA: 9.6%, low Apgar score: 1.3%).

Slightly more than 50% of the mothers lived in a district from one of the two least deprived BIMD quintiles (Q1 and Q2), which were more concentrated in the South of Bavaria, while the more deprived areas were located in the Northeast (Fig 1).

Table 1. Descriptive statistics of all n = 827,105 births between 2009 and 2016 in Bavaria, Germany, with available information on Bavarian Index of Multiple Deprivation (BIMD) at district level. Percentages refer to the total number of non-missing values for each variable.

|  | n (%) |
|---|---|
| Male* | n = 423,952 (51.3%) |
| Twin or higher | n = 30,573 (3.7%) |
| Maternal age > 35 years** | n = 222,748 (26.9%) |
| Gestational diabetes mellitus | n = 32,596 (3.9%) |
| Pre-gestational diabetes mellitus | n = 6,160 (0.7%) |
| Maternal overweight in early pregnancy*** | n = 240,977 (33.0%) |
| Excessive gestational weight gain*** | n = 301,859 (41.4%) |
| Migration background | n = 161,552 (19.5%) |
| Single mother**** | n = 65,687 (9.1%) |
| Multiparous woman | n = 407,682 (49.3%) |
| Smoking during pregnancy | n = 41,294 (5.0%) |
| Substandard use of antenatal care | n = 212,801 (25.7%) |
| Living in a city (>100,000 inhabitants) | n = 224,779 (27.2%) |
| BIMD quintile 1 (least deprived) | n = 204,816 (24.8%) |
| BIMD quintile 2 | n = 260,109 (31.5%) |
| BIMD quintile 3 | n = 115,862 (14.0%) |
| BIMD quintile 4 | n = 116,307 (14.1%) |
| BIMD quintile 5 (most deprived) | n = 130,011 (15.7%) |

* 43 missing values.

** 3 missing values.

*** 97,632 missing values.

**** 101,110 missing values.

In adjusted logistic regression analyses, there was a significantly increased risk for preterm deliveries in mothers from the most deprived compared to the least deprived districts (e.g. OR [95% CI] for Q5 compared to Q1: 1.06 [1.03, 1.09], Table 3), while increased deprivation was associated with lower stillbirth rates (OR [95% CI] per quintile: 0.95 [0.92, 0.98]). A higher BIMD was also associated with lower rates of CS (OR [95% CI] for Q5 compared to Q1: 0.92 [0.90, 0.93], Table 4) and low Apgar scores (OR [95% CI] for Q5 compared to Q1: 0.86 [0.80, 0.94]), but with higher rates of SGA births (OR [95% CI] for Q5 compared to Q1: 1.13 [1.10, 1.16]) in adjusted analyses, respectively. Environment and security deprivation had little or no significant associations with any of the five outcomes, while income, employment, educational, municipal/district revenue, and social capital deprivation were significantly associated with at

Table 2. Prevalences of preterm delivery, stillbirth, Cesarean Sections (CS), Small for Gestational Age (SGA) births, and low Apgar score in n = 827,105 births between 2009 and 2016 in Bavaria, Germany, by Bavarian Index of Multiple Deprivation (BIMD) at district level.

|  | Preterm delivery | Stillbirth | CS | SGA | Low Apgar |
|---|---|---|---|---|---|
| Total | n = 71,772 (8.7%) | n = 2,473 (0.3%) | n = 279,956 (33.9%) | n = 77,020 (9.3%) | n = 10,007 (1.2%) |
| BIMD quintile 1 (least deprived) | n = 18,014 (8.8%) | n = 636 (0.3%) | n = 71,821 (35.1%) | n = 18,214 (8.9%) | n = 2,388 (1.2%) |
| BIMD quintile 2 | n = 21,597 (8.4%) | n = 797 (0.3%) | n = 86,793 (31.0%) | n = 23,570 (9.2%) | n = 3,438 (1.3%) |
| BIMD quintile 3 | n = 10,207 (8.6%) | n = 320 (0.3%) | n = 41,606 (35.0%) | n = 10,908 (9.2%) | n = 1,462 (1.2%) |
| BIMD quintile 4 | n = 10,462 (9.0%) | n = 315 (0.3%) | n = 38,583 (33.2%) | n = 11,074 (9.5%) | n = 1,316 (1.1%) |
| BIMD quintile 5 (most deprived) | n = 11,492 (8.8%) | n = 405 (0.3%) | n = 41,153 (31.7%) | n = 13,254 (10.2%) | n = 1,403 (1.1%) |

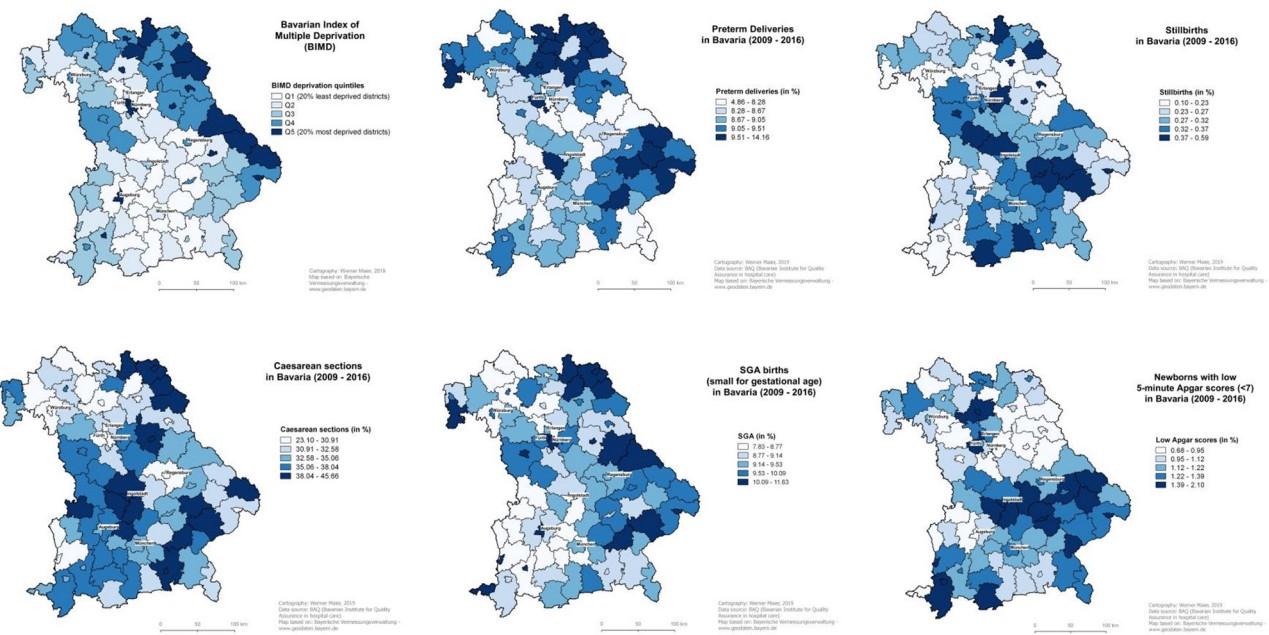

**Fig 1. Maps of Bavaria at district-level of the Bavarian Index of Multiple Deprivation and the percentages of preterm deliveries, stillbirths, Caesarean sections, SGA births and low 5-minute Apgar score in 2009–2016 (cities with >100,000 inhabitants are labelled with their names).**

least four of the five outcomes, respectively (Table 5). This pattern indicates that there was no single domain of the BIMD which was mainly responsible for the observed associations.

The associations between BIMD and Apgar score remained virtually unchanged when additionally adjusted for birth mode (e.g. OR [95% CI] for Q2 compared to Q1: 1.09 [1.02, 1.16], and for Q5 compared to Q1: 0.87 [0.80, 0.94]). We identified one large clinic (>20,000 births in 2009–2016) which had a considerably increased stillbirth rate (> 0.8%) compared to the overall prevalence of stillbirths of (0.3%) in the whole dataset, potentially indicating that this clinic might specialize in terminations of pregnancy. When this clinic was excluded in a posthoc analysis, the ORs of stillbirths by BIMD moved closer to 1 and significant linear trends were no longer observed across BIMD quintiles (S1 Table). The ORs for a primary CS increased with higher BIMD, while the ORs for a secondary CS decreased (S2 Table).

**Table 3. Odds ratios [95% confidence intervals] for preterm delivery (n = 71,772 (8.7%) observations) and stillbirth (n = 2,473 (0.3%) observations) by Bavarian Index of Multiple Deprivation (BIMD) quintiles, unadjusted and adjusted for offspring's sex, multiple delivery, maternal age > 35 years, diabetes during pregnancy, maternal overweight, excessive gestational weight gain, migration background, single mother status, parity, maternal smoking during pregnancy, substandard use of antenatal care, living in a city (>100,000 inhabitants) and year of birth.**

| BIMD | Preterm delivery, crude | Preterm delivery, adjusted | Stillbirth, crude | Stillbirth, adjusted |
|---|---|---|---|---|
| Quintile 1 (least deprived) | Reference | Reference | Reference | Reference |
| Quintile 2 | **0.95 [0.93, 0.97]** | 1.02 [0.99, 1.05] | 1.00 [0.91, 1.11] | 0.91 [0.80, 1.04] |
| Quintile 3 | **0.97 [0.95, 0.995]** | 1.02 [0.99, 1.05] | **0.85 [0.74, 0.98]** | **0.78 [0.67, 0.91]** |
| Quintile 4 | 1.03 [0.999, 1.05] | **1.04 [1.01, 1.08]** | **0.87 [0.76, 0.998]** | **0.79 [0.67, 0.92]** |
| Quintile 5 | 1.01 [0.98, 1.03] | **1.06 [1.03, 1.09]** | 1.00 [0.89, 1.14] | 0.86 [0.74, 1.01] |
| Overall trend (linear increase per quintile) | **1.01 [1.002, 1.01]** | **1.01 [1.01, 1.02]** | 0.98 [0.96, 1.01] | **0.95 [0.92, 0.98]** |

Significant associations (p<0.05) are shown in boldface.

**Table 4. Odds ratios [95% confidence intervals] for Caesarean section (CS; n = 279,956 (33.9%) observations), Small for Gestational Age (SGA) births (n = 77,020 (9.3%) observations) and low Apgar score (n = 10,007 (1.2%) observations) by Bavarian Index of Multiple Deprivation (BIMD) quintiles, unadjusted and adjusted for offspring's sex, multiple delivery, maternal age > 35 years, diabetes during pregnancy, maternal overweight, excessive gestational weight gain, migration background, single mother status, parity, maternal smoking during pregnancy, substandard use of antenatal care, living in a city (>100,000 inhabitants) and year of birth.**

| BIMD | CS, crude | CS, adjusted | SGA, crude | SGA, adjusted | Low Apgar, crude | Low Apgar, adjusted |
|---|---|---|---|---|---|---|
| Quintile 1 (least deprived) | Reference | Reference | Reference | Reference | Reference | Reference |
| Quintile 2 | **0.94 [0.93, 0.95]** | **0.97 [0.95, 0.98]** | **1.03 [1.01, 1.06]** | 1.02 [0.995, 1.05] | **1.15 [1.09, 1.21]** | **1.08 [1.02, 1.16]** |
| Quintile 3 | 1.00 [0.99, 1.02] | **1.05 [1.03, 1.07]** | **1.04 [1.01, 1.06]** | **1.04 [1.01, 1.06]** | 1.06 [0.99, 1.13] | 1.03 [0.95, 1.11] |
| Quintile 4 | **0.92 [0.91, 0.93]** | **0.90 [0.88, 0.91]** | **1.08 [1.05, 1.11]** | **1.05 [1.02, 1.08]** | 0.97 [0.91, 1.04] | 0.93 [0.86, 1.01] |
| Quintile 5 | **0.86 [0.85, 0.87]** | **0.92 [0.90, 0.93]** | **1.16 [1.14, 1.19]** | **1.13 [1.10, 1.16]** | **0.93 [0.87, 0.99]** | **0.86 [0.80, 0.94]** |
| Overall trend (linear increase per quintile) | **0.97 [0.97, 0.97]** | **0.98 [0.98, 0.98]** | **1.03 [1.03, 1.04]** | **1.03 [1.02, 1.03]** | **0.97 [0.96, 0.98]** | **0.96 [0.94, 0.97]** |

Significant associations (p<0.05) are shown in boldface.

**Table 5. Odds ratios [95% confidence intervals] of adverse pregnancy outcomes (CS: Caesarean section; SGA: small for gestational age) per 1 SD increase of Bavarian Index of Multiple Deprivation (BIMD) domains, adjusted for offspring's sex, multiple delivery, maternal age > 35 years, diabetes during pregnancy, maternal overweight, excessive gestational weight gain, migration background, single mother status, parity, maternal smoking during pregnancy, substandard use of antenatal care, living in a city (>100,000 inhabitants) and year of birth.**

| BIMD domain | Preterm delivery | Stillbirth | CS | SGA | Low Apgar |
|---|---|---|---|---|---|
| Income deprivation | **1.01 [1.003, 1.02]** | **0.91 [0.87, 0.96]** | **0.99 [0.99, 0.998]** | **1.04 [1.03, 1.04]** | **0.93 [0.91, 0.95]** |
| Employment deprivation | **1.03 [1.02, 1.04]** | 0.97 [0.92, 1.02] | **0.97 [0.96, 0.97]** | **1.04 [1.03, 1.04]** | **0.93 [0.90, 0.95]** |
| Educational deprivation | **1.02 [1.01, 1.03]** | 0.96 [0.91, 1.002] | **1.01 [1.01, 1.02]** | **1.03 [1.03, 1.04]** | **0.94 [0.92, 0.96]** |
| Municipal/district revenue deprivation | **1.01 [1.003, 1.02]** | 0.97 [0.92, 1.01] | **0.98 [0.97, 0.98]** | **1.03 [1.02, 1.04]** | **0.96 [0.94, 0.98]** |
| Social capital deprivation | 1.01 [0.999, 1.02] | **0.93 [0.89, 0.98]** | **1.01 [1.01, 1.02]** | **1.02 [1.01, 1.03]** | **0.96 [0.94, 0.98]** |
| Environment deprivation | 1.02 [0.993, 1.03] | 0.99 [0.90, 1.08] | **0.89 [0.88, 0.90]** | 1.00 [0.98, 1.02] | 0.98 [0.93, 1.03] |
| Security deprivation | 0.99 [0.98, 1.003] | 1.00 [0.95, 1.05] | 1.00 [0.99, 1.003] | 1.00 [0.99, 1.01] | 1.01 [0.98, 1.03] |

Significant associations (p<0.05) are shown in boldface.

## Discussion

In our study, area-level deprivation was significantly associated with perinatal outcomes, but not always in the expected direction: While the ORs of preterm deliveries, SGA births and primary CS were increased in highly deprived regions, similar to previous studies from the United States [9] and Sweden [11], decreased risks of all other outcomes (stillbirths, secondary CS and low Apgar scores) were observed in highly deprived areas.

This could make sense with respect to CS rates, which are known to be higher in mothers with high SES [29], although the observation of higher primary CS rates (conducted prior to onset of Labour) and of lower secondary CS rates (conducted after the onset of Labour) in highly deprived regions was somewhat contrary to our expectations. We were also surprised about the finding of lower risks of stillbirths and low Apgar scores in deprived Bavarian areas, which is also contrary to previous studies indicating increased risks for both outcomes in other European countries [30, 31]. We do not think that regionally different reporting standards play a major role in this context, in particular not for stillbirths, as this is one of the most important indicators used for obstetric benchmarking and therefore monitored with highest scrutiny. However, we found that one major birth clinic with unusually high stillbirth rates seems to specialize in the termination of pregnancies. Indeed, the observed significant trend of decreased risks of stillbirths in highly deprived areas disappeared when we excluded this clinic

from our analyses. In contrast, we were not able to explain the analogous trend for low Apgar scores by a potential confounding through CS deliveries.

The major strengths of our study are the large size and the high quality of the data analysed. In particular, the completeness and validity of the data are monitored yearly as part of an established national programme of benchmarking health-care provision. Another strength of our study is the use of an area deprivation index based on an established British method [18, 19]. We counted all outcomes for any births and not only for live births in order to avoid healthy-survivor bias.

The fact that our analyses were based on cross-sectional data constitutes a limitation, as we were not able to investigate potential changes in deprivation over time and their impact on pregnancy outcomes. Further, the BIMD quintiles were calculated based on the number of districts not considering the population size of each area. Therefore, the numbers of pregnant women were not of equal size within each BIMD quintile. However, the same issue occurred also in other studies [6, 32], and we do not consider this to be a potential source of bias. Missing covariate data were not evenly distributed across BIMD quintiles and pregnancy outcomes. We therefore cannot preclude that our adjusted analyses were affected by attrition bias, although again we do not think this is a major issue here because the differences in the proportions of missing covariate data were not very large, and the results from the unadjusted and adjusted analyses were quite similar in most cases.

Despite the large sample size, the statistical power of our analyses was somewhat limited due to the relatively low prevalence of most adverse perinatal outcomes. For this reason, we did not correct for multiple testing and consider our analyses as exploratory and our results as hypothesis generating rather than confirmatory.

## Conclusion

We found that area-level deprivation in Bavaria, Germany, was associated with adverse perinatal outcomes. However, while area-level deprivation was positively associated with preterm and SGA births, strengthening other international findings, and also with primary CS, we could not confirm a positive association with area-level deprivation for stillbirth rates and low Apgar scores in this region.

## Supporting information

**S1 Table. Sensitivity analysis for stillbirths after exclusion of a clinic with unusually high rates of stillbirths.** Odds ratios [95% confidence intervals] of stillbirth rates by Bavarian Index of Multiple Deprivation (BIMD) quintiles after exclusion of one major birth clinic with unusually high rates of stillbirths, unadjusted and adjusted for offspring's sex, multiple delivery, maternal age > 35 years, diabetes during pregnancy, maternal overweight, excessive gestational weight gain, migration background, single mother status, parity, maternal smoking during pregnancy, substandard use of antenatal care, living in a city (>100,000 inhabitants) and year of birth. Significant associations (p<0.05) are shown in boldface.
(DOCX)

**S2 Table. Associations with primary and secondary Caesareans Sections (CS).** Odds ratios [95% confidence intervals] for primary (n = 148,225 (17.9%) observations) and secondary CS (n = 131,731 (15.9%) observations) by Bavarian Index of Multiple Deprivation (BIMD) quintiles, unadjusted and adjusted for offspring's sex, multiple delivery, maternal age > 35 years, diabetes during pregnancy, maternal overweight, excessive gestational weight gain, migration background, single mother status, parity, maternal smoking during pregnancy, substandard

use of antenatal care, living in a city (>100,000 inhabitants) and year of birth. Significant associations (p<0.05) are shown in boldface.
(DOCX)

## Author Contributions

**Conceptualization:** Andreas Beyerlein, Werner Maier.

**Data curation:** Nicholas Lack.

**Formal analysis:** Andreas Beyerlein.

**Methodology:** Andreas Beyerlein, Nicholas Lack, Werner Maier.

**Writing – original draft:** Andreas Beyerlein, Werner Maier.

**Writing – review & editing:** Andreas Beyerlein, Nicholas Lack, Werner Maier.

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
