## [Decision Letter · Decision Letter 0]

8 Apr 2020

PONE-D-19-29805

Associations of area-level deprivation with adverse obstetric and perinatal outcomes in Bavaria, Germany: Results from a cross-sectional study

PLOS ONE

Dear Dr. Beyerlein,

Thank you for submitting your manuscript to PLOS ONE. After careful consideration, we feel that it has merit but does not fully meet PLOS ONE’s publication criteria as it currently stands. Therefore, we invite you to submit a revised version of the manuscript that addresses the points raised during the review process.

We would appreciate receiving your revised manuscript by May 23 2020 11:59PM. To enhance the reproducibility of your results, we recommend that if applicable you deposit your laboratory protocols in protocols.io, where a protocol can be assigned its own identifier (DOI) such that it can be cited independently in the future. For instructions see: http://journals.plos.org/plosone/s/submission-guidelines#loc-laboratory-protocols

We look forward to receiving your revised manuscript.

Kind regards,

Kannan Navaneetham

Academic Editor

PLOS ONE

Journal Requirements:

2. Thank you for submitting the above manuscript to PLOS ONE. During our internal evaluation of the manuscript, we found significant text overlap between your submission (mainly within the discussion section) and the following previously published work, on which you are an author.

https://bmcpregnancychildbirth.biomedcentral.com/articles/10.1186/s12884-016-1060-3

Please revise the manuscript to rephrase the duplicated text, cite your sources, and provide details as to how the current manuscript advances on previous work. Please note that further consideration is dependent on the submission of a manuscript that addresses these concerns about the overlap in text with published work.

Reviewers' comments:

Reviewer's Responses to Questions

**Comments to the Author**

1. Is the manuscript technically sound, and do the data support the conclusions?

Reviewer #1: Partly

2. Has the statistical analysis been performed appropriately and rigorously? 

Reviewer #1: Yes

3. Have the authors made all data underlying the findings in their manuscript fully available?

Reviewer #1: Yes

4. Is the manuscript presented in an intelligible fashion and written in standard English?

Reviewer #1: Yes

5. Review Comments to the Author

Reviewer #1: Associations of area-level deprivation with adverse obstetric and perinatal outcomes in Bavaria, Germany: Results from a cross-sectional study is a well written, important and interesting manuscript. The aim of the study is clear and the conclusions are well fit to their findings and any limitations were also taken into account.

My main point of criticism is that the description of the data used in the analyses is very limited and used definitions are not always clear.

Points for revision:

abstract: -

background:

lines 63-65: neighborhood deprivation indices not only serve as an alternative when individual level SES metrics are not available, but also capture other aspects of deprivation, which are often missed when only looked at individual level metrics. It also captures local availability of care and other area specific care and policy structures. The authors should elaborate more on these assets of area measures

line 67: introduce abbreviation BIMD here instead of in method section

methods:

How is the BIMD calculated? on a multiplicative or additive scale? is it standardized? is this only available for Bavaria? or also for the rest of Germany (and how does the level of deprivation in Bavaria compare to Germany)?

How do the individual domains relate to the composite? and why only look for a linear trend over the domains and not like the main analyses (line 98)?

line 87: what is the size (plus range) of a district?

line 91: "We used data on all deliveries..." where there any restrictions to be included in the dataset? like minimal amount of weeks of gestation of minimal birth weight? Viability? This has considerable implications for the interpretation of your findings.

How were the stillbirths defined?

Was there a distinction possible between primary and secondary caesareans? This distinction should be made since their role in the management of deliveries is very different. The associations with deprivation of both types could well be different and capture different aspects of inequality in care provision

lines 97-98: which analyses do you refer to here? The trend over the quintiles? It is not entirely clear to me which results match these analyses

why did you choose to correct for multiple pregnancies instead of stratifying? Were the number of multiple pregnancies comparable over deprivation quintiles? most studies limit their analyses to singletons only and now it is more difficult to directly compare your results with that of others.

did you have information on other birth modes than CS only? if yes, why not correct for that?

how did you handle missing data? how was missing covariate data distributed among the deprivation quintiles and outcomes?

were there specific rules in place to define unusually high rates of stillbirths? or was it a post-hoc decision to see whether the outlier clinic influenced your findings? If so, than please state that... the current sentence is very vague about this point/decision

results:

I would like to see a table displaying the prevalence of each outcome, for the whole population and over the quintiles

Tables 2-4: please provide additional information about the N for analyses, per outcome and per quintile. This is especially of interest since 'single mother' is a covariate with a large amount of missing data, and this covariate is likely not evenly distributed among the quintiles, potentially introducing bias

lines 129-136: this paragraph on the different domains was hard to read, not sue what the message here is

line 138: please provide ORs + 95%CIs for these analyses instead of "data not shown"

line 139: what was the size of the excluded clinic? how many births were excluded from the analyses?

discussion:

lines 151-153: this may be influenced by different definitions; maybe the increased amount of perinatal death in other studies focus on the period just after birth, while yours could just focus on IFD? or other mechanisms... Also, what were the unadjusted estimates for this outcome? do they differ? on the other hand, your amount of stillbirths is low, which could influence your findings. also, selective reporting of stillbirths in the database could have played a role in this. Please take this into account in discussing your findings on stillbirths in particular.

lines 159-161: were the apgar scores (but also SGA and preterm birth) only counted for live births or any births? This could have influenced the found associations (similar to healthy survivor bias) and should be addressed

lines 170-172: similar to my suggestion in the background; area SES is not just an 'alternative' to use when individual level SES is absent, it captures other aspects of SES and also is also associated with health outcomes, irrespective of individual level SES (like in the study of Daoud BMC Pregnancy Childbirth. 2015)

I miss the limitation that it was a cross-sectional study; especially since there is ample evidence suggestion that SES is not a fixed thing, but evolves over time and the effect of SES differs with different degrees and lengths of exposure

6. PLOS authors have the option to publish the peer review history of their article (what does this mean?). If published, this will include your full peer review and any attached files.

Reviewer #1: No

---

## [Author Response · Author response to Decision Letter 0]

29 May 2020

Dear Dr. Navaneetham,

thank you for considering our paper for publication in PLOS ONE. We appreciate the constructive comments of the reviewers and have changed the manuscript according to their suggestions, using highlighting mode. We have answered the comments as follows:

Journal Requirements:

Thank you for pointing this out to us. We have adjusted the manuscript layout accordingly.

2. Thank you for submitting the above manuscript to PLOS ONE. During our internal evaluation of the manuscript, we found significant text overlap between your submission (mainly within the discussion section) and the following previously published work, on which you are an author.

https://bmcpregnancychildbirth.biomedcentral.com/articles/10.1186/s12884-016-1060-3

Please revise the manuscript to rephrase the duplicated text, cite your sources, and provide details as to how the current manuscript advances on previous work. Please note that further consideration is dependent on the submission of a manuscript that addresses these concerns about the overlap in text with published work.

Thank you for this hint. We have revised the text accordingly.

In order to make de-identified data available for public use prior permission must be obtained from the Bavarian institute for quality assurance (‘BAQ’). The reason is that the data is expressly gathered for purposes of quality improvement of obstetrics and neonatology care. Hence, the BAQ steering committee must assert that any request for third party usage of the data is in line with the general BAQ objectives. The BAQ is bound by corresponding contracts with its participating hospitals to guarantee adherence to this protocol

Thank you for pointing this out. We have replaced „data not shown“ by specific odds ratios now.

Thank you very much, we have changed this accordingly.

Review Comments to the Author

Reviewer #1: Associations of area-level deprivation with adverse obstetric and perinatal outcomes in Bavaria, Germany: Results from a cross-sectional study is a well written, important and interesting manuscript. The aim of the study is clear and the conclusions are well fit to their findings and any limitations were also taken into account.

My main point of criticism is that the description of the data used in the analyses is very limited and used definitions are not always clear.

lines 63-65: neighborhood deprivation indices not only serve as an alternative when individual level SES metrics are not available, but also capture other aspects of deprivation, which are often missed when only looked at individual level metrics. It also captures local availability of care and other area specific care and policy structures. The authors should elaborate more on these assets of area measures

We thank the reviewer for these important indications and have changed the text accordingly:

“Area-level deprivation research may consider the proportion of deprived people in an area (‘compositional meaning’), the presence of possible area effects beyond the local concentration of deprived people (’collective meaning’), and the lack of infrastructural facilities or other area features (‘environmental meaning’) [15]. Area-level deprivation is often measured with standardised composite deprivation indices describing a structural lack of material and social resources in an area, considering e. g. income and occupation but also municipal/district revenues and environmental indicators. Therefore, these indices are valuable instruments for the analysis of area effects on health beyond the individual level, even when they might be used as a surrogate when individual socioeconomic data are not available [16].”

line 67: introduce abbreviation BIMD here instead of in method section

Thank you very much, we have changed this accordingly.

How is the BIMD calculated? on a multiplicative or additive scale? is it standardized? is this only available for Bavaria? or also for the rest of Germany (and how does the level of deprivation in Bavaria compare to Germany)?

Thank you very much, we acknowledge the need to integrate more information on the BIMD in the text and have added following information in the background section:

“Based on an established British method [15], the BIMD is a multidimensional construct combining standardised and transformed indicators in specific deprivation domains which are weighted and combined in an overall deprivation index. The BIMD was originally developed as a model for the development of the German Index of Multiple Deprivation (GIMD), being today part of a number of Indices of Multiple Deprivation (IMD) for Germany and some of its federal states. More on the methodology of the German IMD, the area-level distribution of deprivation in Germany and its association with health can be found elsewhere [17-20].”

How do the individual domains relate to the composite? and why only look for a linear trend over the domains and not like the main analyses (line 98)?

The individual domains load with a certain amount on the composite score. We have added these amounts (in %) to the main text. We decided to investigate them with a linear trend only in order to reduce multiple testing issues.

line 87: what is the size (plus range) of a district?

By the reporting date of 31 December 2016, the smallest district in Bavaria had a population of 40.7 thousand inhabitants (urban district of Schwabach), whereas the largest district (city of Munich) had a population of 1.46 million inhabitants (range: 1.42 million). The median number of inhabitants was 115.6 thousand. We have now inserted additional information on the size of the districts in the text. 

line 91: "We used data on all deliveries..." where there any restrictions to be included in the dataset? like minimal amount of weeks of gestation of minimal birth weight? Viability? This has considerable implications for the interpretation of your findings.

No, we deliberately applied no inclusion restrictions on our data. 

How were the stillbirths defined?

All neonates without signs of life and minimum birth weight of 500 g where counted as stillbirths.

Was there a distinction possible between primary and secondary caesareans? This distinction should be made since their role in the management of deliveries is very different. The associations with deprivation of both types could well be different and capture different aspects of inequality in care provision

Thank you for this interesting suggestion. We added a supplementary analysis for primary and secondary sectio (S2 Table) and commented on it in the main text.

lines 97-98: which analyses do you refer to here? The trend over the quintiles? It is not entirely clear to me which results match these analyses

Yes, this refers to the ‚overall trend‘ analyses. We have added this term for clarity.

why did you choose to correct for multiple pregnancies instead of stratifying? Were the number of multiple pregnancies comparable over deprivation quintiles? most studies limit their analyses to singletons only and now it is more difficult to directly compare your results with that of others.

There were two reasons why we did not stratify for multiple pregnancies: First, this would have doubled the number of analyses and hence contributed to multiple testing issues. Second, the statistical power in the subset of multiple pregnancies would likely have been too low to draw valid conclusions. 

did you have information on other birth modes than CS only? if yes, why not correct for that?

Thank you. The only other birth mode available in addition to vaginal birth and CS was vaginal surgery. We added this variable to the sensitivity analysis for Apgar scores.

how did you handle missing data? how was missing covariate data distributed among the deprivation quintiles and outcomes?

We added the following to the Results and Discussion, respectively: 

„Missing covariate data occurred for n=28,419 observations (13.9 %) in Q1, n=45,553 (17.7 %) in Q2, n=23,108 (19.4 %) in Q3, n=24,634 (21.2 %) in Q4 and n=26,301 (20.2 %) in Q5, respectively. In pregnancies with missing covariate data, slightly higher prevalences were observed for all outcomes (preterm delivery: 9.1 %, stillbirth: 0.4 %, CS: 35.9 %, SGA: 9.6 %, low Apgar score: 1.3 %).“

„Missing covariate data were not evenly distributed across BIMD quintiles and pregnancy outcomes. We therefore cannot preclude that our adjusted analyses were affected by attrition bias, although again we do not think this is a major issue here because the differences in the proportions of missing covariate data were not very large, and the results from the unadjusted and adjusted analyses were quite similar in most cases.“

were there specific rules in place to define unusually high rates of stillbirths? or was it a post-hoc decision to see whether the outlier clinic influenced your findings? If so, than please state that... the current sentence is very vague about this point/decision

Thank you. We have added that this was a posthoc decision.

I would like to see a table displaying the prevalence of each outcome, for the whole population and over the quintiles

Thank you for this suggestion. We have added this table as new table 2.

Tables 2-4: please provide additional information about the N for analyses, per outcome and per quintile. This is especially of interest since 'single mother' is a covariate with a large amount of missing data, and this covariate is likely not evenly distributed among the quintiles, potentially introducing bias

This issue has already been addressed by our answer to your other question related to missing covariates further above: As there were no missing values for any of the outcome variables, the N per quintile was the same for each adjusted analysis, irrespectively of the outcome variable. 

lines 129-136: this paragraph on the different domains was hard to read, not sue what the message here is

Thank you. We have revised this paragraph and hope that it reads clearer now.

line 138: please provide ORs + 95%CIs for these analyses instead of "data not shown"

Thank you. We followed your advice.

line 139: what was the size of the excluded clinic? how many births were excluded from the analyses?

Due to data protection reasons, we cannot mention the exact size of this clinic. We added the information that this clinic had >20,000 births in 2009-2016.

lines 151-153: this may be influenced by different definitions; maybe the increased amount of perinatal death in other studies focus on the period just after birth, while yours could just focus on IFD? or other mechanisms... Also, what were the unadjusted estimates for this outcome? do they differ? on the other hand, your amount of stillbirths is low, which could influence your findings. also, selective reporting of stillbirths in the database could have played a role in this. Please take this into account in discussing your findings on stillbirths in particular.

Thank you for addressing this issue. The unadjusted and adjusted ORs for stillbirths were relatively similar, as shown in table 3. Selective reporting of stillbirths is unlikely, because stillbirths are a key outcome for benchmarking of birth clinics and is therefore rigorously checked during data collection, as was already mentioned in the discussion. 

lines 159-161: were the apgar scores (but also SGA and preterm birth) only counted for live births or any births? This could have influenced the found associations (similar to healthy survivor bias) and should be addressed

All outcomes were counted for any births. We added one sentence about this to the Discussion.

lines 170-172: similar to my suggestion in the background; area SES is not just an 'alternative' to use when individual level SES is absent, it captures other aspects of SES and also is also associated with health outcomes, irrespective of individual level SES (like in the study of Daoud BMC Pregnancy Childbirth. 2015)

Thank you. We have removed the respective sentence.

I miss the limitation that it was a cross-sectional study; especially since there is ample evidence suggestion that SES is not a fixed thing, but evolves over time and the effect of SES differs with different degrees and lengths of exposure.

Thank you. We have added the following sentence: „The fact that our analyses were based on cross-sectional data constitutes a limitation, as we were not able to investigate potential changes in deprivation over time and their impact on pregnancy outcomes.“

We hope that all objections of the Reviewer could be sufficiently addressed and would be pleased if the amended manuscript would be accepted for publication.

Yours sincerely,

Andreas Beyerlein (for all authors)

---

## [Decision Letter · Decision Letter 1]

19 Jun 2020

PONE-D-19-29805R1

Associations of area-level deprivation with adverse obstetric and perinatal outcomes in Bavaria, Germany: Results from a cross-sectional study

PLOS ONE

Dear Dr. Beyerlein,

Thank you for submitting your manuscript to PLOS ONE. After careful consideration, we feel that it has merit but does not fully meet PLOS ONE’s publication criteria as it currently stands. Therefore, we invite you to submit a revised version of the manuscript that addresses the points raised during the review process.

We look forward to receiving your revised manuscript.

Kind regards,

Kannan Navaneetham, PhD

Academic Editor

PLOS ONE

Reviewers' comments:

Reviewer's Responses to Questions

**Comments to the Author**

1. If the authors have adequately addressed your comments raised in a previous round of review and you feel that this manuscript is now acceptable for publication, you may indicate that here to bypass the “Comments to the Author” section, enter your conflict of interest statement in the “Confidential to Editor” section, and submit your "Accept" recommendation.

Reviewer #1: (No Response)

2. Is the manuscript technically sound, and do the data support the conclusions?

Reviewer #1: Yes

3. Has the statistical analysis been performed appropriately and rigorously? 

Reviewer #1: Yes

4. Have the authors made all data underlying the findings in their manuscript fully available?

Reviewer #1: No

5. Is the manuscript presented in an intelligible fashion and written in standard English?

Reviewer #1: Yes

6. Review Comments to the Author

Reviewer #1: The vast majority of the comments/concerns raised have been successfully covered by the authors.

I have two minor but important points that should be changed in the manuscript.

1. definition of stillbirths: please include the used definition in the methods section. Also, you described in the rebuttal, that all neonates without signs of life and minimum birth weight of 500 g where counted as stillbirths. but since you have no restrictions for the population used in the analyses, how do you deal with any birth with a birth weight below 500gr? if they are excluded, than please add that to the description of your population, if they were dealt with in another way, plase describe this in the ethods section

2. internationally, the term primary CS is use dto describe the elective or planned CS, and secundary CS those following complications when labour had already started. in your discussion section you use the terms the other way around. please correct this.

7. PLOS authors have the option to publish the peer review history of their article (what does this mean?). If published, this will include your full peer review and any attached files.

Reviewer #1: No

---

## [Author Response · Author response to Decision Letter 1]

25 Jun 2020

Dear Dr. Navaneetham,

thank you for considering our paper for publication in PLOS ONE. We have answered the Reviewer’s remaining comments as follows:

Reviewer #1: The vast majority of the comments/concerns raised have been successfully covered by the authors.

I have two minor but important points that should be changed in the manuscript.

1. definition of stillbirths: please include the used definition in the methods section. Also, you described in the rebuttal, that all neonates without signs of life and minimum birth weight of 500 g where counted as stillbirths. but since you have no restrictions for the population used in the analyses, how do you deal with any birth with a birth weight below 500gr? if they are excluded, than please add that to the description of your population, if they were dealt with in another way, plase describe this in the ethods section

Thank you for raising this issue. Stillborn neonates below a birth weight of 500 g are not required for official registration in Germany and hence they are also not part of the database. This means that neonates with a birth weight below 500 g enter the database only if they show signs of life, but not if they are stillbirths. We have clarified this in the main text. 

2. internationally, the term primary CS is use dto describe the elective or planned CS, and secundary CS those following complications when labour had already started. in your discussion section you use the terms the other way around. please correct this. 

The distinction between primary and secondary Caesarean sections is in fact - contrary to your assumption – a German rather than an International Convention. Internationally the distinction is generally between elective and emergency Caesarean sections. However, when referring to primary and secondary Caesarean sections as we did in our article we clearly intend to use the term primary for such Caesarean sections when conducted prior to onset of Labour as distinguished from so called secondary Caesarean sections which are defined as Caesarean sections conducted after the onset of Labour. We apologize for the confusion and have revised this part of the Discussion accordingly.

We hope that all objections of the Reviewer could be sufficiently addressed and would be pleased if the amended manuscript would be accepted for publication.

Yours sincerely,

Andreas Beyerlein (for all authors)

---

## [Editor Report · Decision Letter 2]

29 Jun 2020

Associations of area-level deprivation with adverse obstetric and perinatal outcomes in Bavaria, Germany: Results from a cross-sectional study

PONE-D-19-29805R2

Dear Dr. Beyerlein,

We’re pleased to inform you that your manuscript has been judged scientifically suitable for publication and will be formally accepted for publication once it meets all outstanding technical requirements.

Kind regards,

Kannan Navaneetham, PhD

Academic Editor

PLOS ONE
---

## [Editor Report · Acceptance letter]

7 Jul 2020

PONE-D-19-29805R2 

Associations of area-level deprivation with adverse obstetric and perinatal outcomes in Bavaria, Germany: Results from a cross-sectional study 

Dear Dr. Beyerlein:

I'm pleased to inform you that your manuscript has been deemed suitable for publication in PLOS ONE. Congratulations! Your manuscript is now with our production department. 

Kind regards, 

on behalf of

Professor Kannan Navaneetham 

Academic Editor

PLOS ONE